# Understanding stakeholder perspectives on Apgar score, cyanosis and identifying jaundice in ethnic minority neonates

**Frankie Fair[1], Amy Furness [1], Gina Higginbottom[2], Sam Oddie [3], Hora Soltani [1] ***

**1** Sheffield Hallam University, Sheffield, United Kingdom, **2** University of Nottingham, Nottingham, United Kingdom, **3** Bradford Teaching Hospitals NHS Foundation Trust, Bradford, United Kingdom

\* h.soltani@shu.ac.uk

## Abstract

### Objectives

To explore neonatal assessments that include an element of evaluating skin colour in neonates of black, Asian and minority ethnicities, focusing on the Apgar score, presence of cyanosis and presence of jaundice.

### Design

We employed focused ethnography involving diverse healthcare professionals (HCPs) and parents or caregivers with Black, Asian, or ethnic minority children born in the last five years. Zoom interviews were performed following a semi-structured interview guide. Ethnographic data analysis was conducted using NVivo software.

### Results

There was a lack of consistency in how HCPs applied the Apgar scores, which also varied from textbook guidelines. The use of pink as a skin colour descriptor for ethnic minority neonates created a divide among both parents and HCPs. HCPs relied more heavily on other elements of the Apgar score or infant breathing and crying at birth to gauge infant wellness. When identifying cyanosis, HCPs depend on specific body locations for accurate assessment of oxygenation, but the limitations of visual assessment are acknowledged. For jaundice, most HCPs recognised the difficulty in identifying the colour yellow in infants with darker skin tones. HCPs focussed on yellowing of the sclera and gums and other well-being assessments to detect jaundice. Some interviewed parents noticed jaundice in their infants before HCPs but felt they were not listened to when raising concerns. HCPs acknowledged the need for additional training to effectively assess the health of ethnic minority infants.

### Conclusion

This study highlights disparities in neonatal health assessment from the perspectives of healthcare professionals and parents from diverse backgrounds. Varied practices in determining the Apgar score and recognising cyanosis and jaundice signal the need for

**Data Availability Statement:** All relevant data are within the manuscript and its Supporting information files.

**Funding:** The research was commissioned by the NHS Race and Health Observatory. The funders had no role in data collection and analysis, decision to publish, or preparation of the manuscript.

**Competing interests:** The authors have declared that no competing interests exist.

standardised protocols, appropriate educational materials, and targeted training. Addressing these challenges is vital for equitable care, emphasising comprehensive training and bias mitigation efforts in maternal and neonatal healthcare.

## Introduction

Ethnic and racial disparities in maternal and neonatal health in the UK are well documented [1]. Compared with White women, Black and Asian women have higher maternal and neonatal mortality and morbidity rates [1, 2]. Disparities in maternity care quality for ethnic minorities have been reported [3–5]. Concerns exist about perinatal procedures potentially being biased towards White infants, disadvantaging darker-skinned infants [6].

Skin tone may influence neonatal assessments, resulting in possible observer and racial bias. One such neonatal assessment is the Apgar score, a tool widely used for over 70 years to determine neonatal well-being at birth [7] which has been primarily validated in white babies [8]. The Apgar score involves five components, one of which is "appearance", involving skin colour [9, 10]. There are concerns about the applicability of the appearance component to ethnic minority neonates [10], as it includes assessing whether the neonate is 'pink all over' [11].

Cyanosis is primarily observed by skin colour assessment; however, its ability to detect cyanosis visually is controversial, especially in darker-skinned infants [12, 13]. Terminology, such as "going blue" for cyanosis or hypoxia in ethnic minority patients, has been questioned, with skin colour alone deemed insufficient to determine oxygenation [14]. Therefore, pulse oximetry screening is recommended [15]. However, pulse oximetry is traditionally calibrated for White skin [16] and has been noted to result in a racial bias that may endanger Black patients [17–19].

Jaundice, marked by yellowing of the skin and sclera, reflects hyperbilirubinemia [20]. Screening for hyperbilirubinemia often initially relies on visual assessment of neonatal skin colour; however, this method is prone to inaccuracy due to pigmentation [21, 22]. The consequences of missed jaundice can profoundly impact infants and their families, potentially leading to kernicterus, a condition shown to disproportionately affect non-Caucasian infants [23, 24]. Transcutaneous bilirubin (TCB) devices have been developed to screen for jaundice to reduce blood sampling in neonates [25, 26]. However, concerns persist about racial bias in TCB diagnosis, overestimating bilirubin in darker-skinned neonates [27, 28].

There is minimal literature that investigates the experiences of healthcare professionals (HCPs) when assessing infants for cyanosis, jaundice, and the Apgar score, particularly in infants with darker or different skin colours [29]. Furthermore, this scarcity in the literature extends to parental experiences of receiving care for infants with darker or different skin colours for jaundice and cyanosis.

Therefore, this study aimed to explore stakeholders' views of the Apgar score, detection of cyanosis and detection of jaundice, with a particular focus on Black, Asian and minority ethnic neonates.

## Methods

### Methodology

A focused ethnography approach was used. Focused ethnography is recognised as learning "about people by learning from them" [30]. It is also a pragmatic way to gather data and

determine areas where care can be enhanced [31]. This approach has increasingly been used to understand and improve practice, as focused ethnography provides an edge over traditional methods by delving into specific community aspects, facilitating deeper analysis within targeted areas of interest [32].

## Participants

A range of stakeholders, including HCPs and parents, were purposively sampled. Eligible HCPs included midwives, obstetricians, health visitors, neonatologists, paediatricians, or neonatal nurses who worked with neonates from Black, Asian or ethnic minority backgrounds. Parents or carers were eligible for inclusion within the study if they cared for a neonate from a Black, Asian or ethnic minority background within the last five years.

Posters were used to advertise the study through professional organisations, local networks, LinkedIn, Facebook and Twitter. Interested participants were requested to contact the research team. Maximum variation within the sample was ensured by recruiting HCPs from various professions, at various stages of their career, and who were of different ethnicities. Additionally, HCPs were recruited from workplaces in areas with both high and low ethnic diversity. Parents and carers were also screened before the interviews were arranged to ensure demographic, geographical, and ethnic diversity among the participants. Recruitment was undertaken between 31st August 2022 and 27th January 2023. Recruitment continued until no new concepts emerged from the data, so data saturation was felt to have been achieved.

## Data collection

A semi-structured interview schedule was used. This included demographic questions and open-ended inquiries to gather in-depth responses. HCPs were asked to share their experiences assessing Apgar scores, cyanosis, and jaundice in ethnic minority neonates. Parents were asked about their experiences seeking care for their ethnic minority neonates, particularly regarding concerns about oxygenation or jaundice. The comprehensiveness, acceptability, and clarity of the interview schedule were confirmed through stakeholder collaboration and pilot testing with two HCPs and two parents. To enable wide geographical participation, interviews were conducted via Zoom. Previous research has shown minimal impact on the number of codes assigned within the analysis to video interviews compared to face-to-face interviews [33]. All interviews were recorded and transcribed verbatim.

## Data analysis

Qualitative data analysis followed Roper and Shapira's (2000) systematic approach [30]. The steps included becoming familiar with the data, line-by-line coding, pattern grouping, outlier identification, construct formation, and memoing reflections on the researcher's emerging thinking. To ensure credibility, independent coding of all transcripts was undertaken by two researchers, with a subset also coded by others within the research team. Following this, the whole team discussed and agreed on the patterns and constructs that emerged. Trustworthiness was reinforced through member checking, participant validation, and three stakeholder workshops. The ethnic and professional diversity of the research team and stakeholders strengthened the process. A narrative thematic analysis is supported by direct quotations to confirm the researcher's interpretation. NVivo software was utilised for the analysis.

## Ethical considerations

This study received ethics approval from Sheffield Hallam University, [ER44006021]].

All participants provided voluntary informed consent to participate in the study. Before participation, participants were informed of the voluntary nature of their involvement, and their right to withdraw at any time. They were assured of the confidentiality and anonymity of their responses, data was stored securely and accessed only by authorised personnel.

By participating in the study, participants consented to the use of their anonymised data for research purposes and publication, with pseudonyms used within the publication.

## Results

### Demographics

Thirty-three HCPs were interviewed, with an average of 46 minutes per interview. As presented in Table 1, the majority were female, with diverse roles: thirteen midwives, eight health visitors, four paediatricians, three obstetricians, three neonatologists, and two neonatal nurses. Eleven HCPs were identified as Black, three as Mixed ethnicity, two as Asian and fifteen as White British. Some HCPs held qualifications from non-UK countries, including Ghana, Greece, Jamaica and South Africa. Thirteen HCPs had practised for ten years or less, eight for 11–20 years, and twelve for more than 20 years. Their work spanned various locations across the UK, as well as internationally, including Africa and Asia.

The key characteristics of the participating parents are presented in Table 2 and in Figs 1 and 2. Twenty-four parents were interviewed; eleven were born in the UK, while 13 were born elsewhere, including Ghana (n = 4), Kenya (n = 1), the Netherlands (n = 1), South Africa (n = 4), and Sri Lanka (n = 1). Two participants did not specify the country stating Africa (n = 1) or Western Africa (n = 1). The majority had a graduate or postgraduate education (79%) (Fig 1). All were employed (Fig 2), with fourteen (58%) in managerial or professional roles, six (25%) in intermediate positions, and four (17%) in routine or manual jobs.

Within the Tables 1 and 2, participants are referred to as the overall category of black, Asian, mixed, other or white to protect confidentiality given the limited numbers of participants within further subcategories of ethnicity.

### Skin colour assessments

Three neonatal assessments are explored below, including how each was assessed in practice and whether there were any specific considerations or challenges in the assessment of ethnic minority neonates.

### Apgar scores

**Differences in pigmentation at birth.**   During the interview, nine HCPs and three of the parents spoke about differences in infant skin colour at birth, recognising that pigmentation developed and changed over the first few days.

*"When a Black baby's first born, they're not always, their pigment is not always fully developed then, so you can see clearly."*

(MW07 –Black)

**Assessment in practice.**   After being presented with the official way to assess an Apgar score, many HCPs were shocked at the terminology used, and they suggested that it was a practical tool based on experience rather than a strict guideline. Healthy babies often received automatic scores of 9, especially if they were breathing normally, while other HCPs described

**Table 1. Demographics of the healthcare provider interviewees.**

| Participant | Gender | Race | Years in practice | Estimated proportion of Black, Asian, and minority ethnic infants in their service | Country of primary qualification |
|---|---|---|---|---|---|
| MW01 | F | Mixed | ≤5 | 25% | UK |
| MW02 | F | Black | 11–20 | 60% | UK |
| MW03 | F | Black | >25 | 40% | UK |
| MW04 | F | Asian | 11–20 | 32% | UK |
| MW05 | F | Mixed | 6–10 | 90% | UK |
| MW06 | F | Black | ≤5 | 65–70% | UK |
| MW07 | F | Black | 21–25 | 30% | UK |
| MW08 | F | White | 6–10 | 40% | UK |
| MW09 | F | White | ≤5 | 2% | UK |
| MW10 | F | White | 11–20 | 90% | Non-UK |
| MW11 | F | White | 6–10 | 38–40% | UK |
| MW12 | F | White | 6–10 | 25%. | UK |
| MW13 | F | White | >25 | 50% | UK |
| NNP01 | M | Black | 11–20 | 100% | Non-UK |
| NNP02 | F | Black | ≤5 | 50% | Non-UK |
| NNP03 | F | White | 11–20 | 50% | UK |
| NNP04 | M | Mixed | 11–20 | 50% | UK |
| NNP05 | F | White | ≤5 | 10% | UK |
| NNP06 | F | White | ≤5 | 1–5% | UK |
| NNP07 | M | White | >25 | 50% | Non-UK |
| NNP08 | F | White | >25 | 30–40% | UK |
| NNP09 | F | Black | 6–10 | 40% | Non-UK |
| OB01 | M | White | >25 | <1% Location 1<br>20% Location 2 | UK |
| OB02 | F | Black | 6–10 | 40% Location 1<br>3% Location 2 | UK |
| OB03 | M | Asian | >25 | 61% | UK |
| HV01 | F | White | >25 | 98% | UK |
| HV02 | F | Black | 21–25 | 60% | UK |
| HV03 | F | White | 21–25 | 25% | UK |
| HV04 | F | White | 11–20 | 5% | UK |
| HV05 | F | Black | 6–10 | 40% | UK |
| HV06 | F | White | >25 | 50% | UK |
| HV07 | F | White | 21–25 | 20% | UK |
| HV8 | F | Black | 11–20 | 65–70% | UK |

MW = Midwife; HV = Health visitor; OB = obstetrician; NNP = neonatologist, neonatal nurse or paediatrician

assigning the Apgar score in retrospect. The Apgar score was viewed as beneficial in education and for assessing the need for resuscitation, although some HCPs viewed the pragmatic use of the Apgar score as crude and questioned its clinical value. There were variations in practice, with some HCPs using parental complexion to aid in colour assessment, while others deemed this inaccurate due to mixed-race family variations.

Regarding the relative importance of individual Apgar score components, opinions varied among HCPs. Some participants felt that the appearance component was subjective, while others prioritised other components, such as heart rate, over appearance.

**Table 2. Demographics of the parent interviewees.**

| Participant code | Number of children | Relation to child | Race | Age | Country of Birth |
|---|---|---|---|---|---|
| PA01 | 1 | Mother | Other | 25–29 | UK |
| PA02 | 1 | Mother | Asian | ≥35 | Non-UK |
| PA03 | 2 | Father | Mixed | 30–34 | Non-UK |
| PA04 | 3 | Mother | Black | 30–34 | UK |
| PA05 | 1 | Father | Black | 25–29 | Non-UK |
| PA06 | 2 | Mother | Black | 30–34 | Non-UK |
| PA07 | 2 | Mother | Black | 30–34 | Non-UK |
| PA08 | 2 | Mother | Black | 25–29 | Non-UK |
| PA09 | 2 | Mother | Black | 25–29 | Non-UK |
| PA10 | 1 | Mother | Black | 30–34 | Non-UK |
| PA11 | 2 | Father | Black | 30–34 | Non-UK |
| PA12 | 1 | Mother | Black | 25–29 | Non-UK |
| PA13 | 2 | Mother | Asian | ≥35 | UK |
| PA14 | 1 | Mother | White | 30–34 | UK |
| PA15 | 1 | Mother | Black | 25–29 | UK |
| PA16 | 2 | Mother | Asian | ≥35 | UK |
| PA17 | 3 | Mother | Black | ≥35 | Non-UK |
| PA18 | 1 | Mother | Black | 30–34 | Non-UK |
| PA19 | 2 | Mother | Black | 30–34 | Non-UK |
| PA20 | 1 | Mother | Black | ≥35 | UK |
| PA21 | 1 | Mother | Other | 25–29 | UK |
| PA22 | 1 | Mother | Mixed | 30–34 | UK |
| PA23 | 3 | Mother | Black | ≥35 | UK |
| PA24 | 1 | Father | Mixed | 30–34 | UK |

*"The fundamental you need to know is breathing or not. What's the heart rate doing?"*

(NNP03 –White)

Some midwives emphasised the importance of recognising central colour changes in different ethnicities and believed it should be evident to any HCP assessing properly.

Given the individual variation in practice, several HCPs recognised that their practice may differ from that of other HCPs, particularly regarding thresholds for action. Some expressed uncertainty about how others assessed the Apgar score and believed it would be assessed differently in countries with a majority ethnic minority population.

*"I don't know what midwives do, whether they look at the soles of the feet or the palms of the hands, or what they do, whether they just think this baby's fine, they'll just tick the number 2 on the appearance."*

(NNP08 –White)

**Relevance to ethnic minority neonates.** HCPs suggested that the Apgar score terminology was originally designed for White infants, which posed challenges for ethnic minority neonates. Some HCPs questioned the use of "*pink*" to assess neonatal well-being, expressing concerns about its applicability to black or Asian babies. In addition, some parents felt that

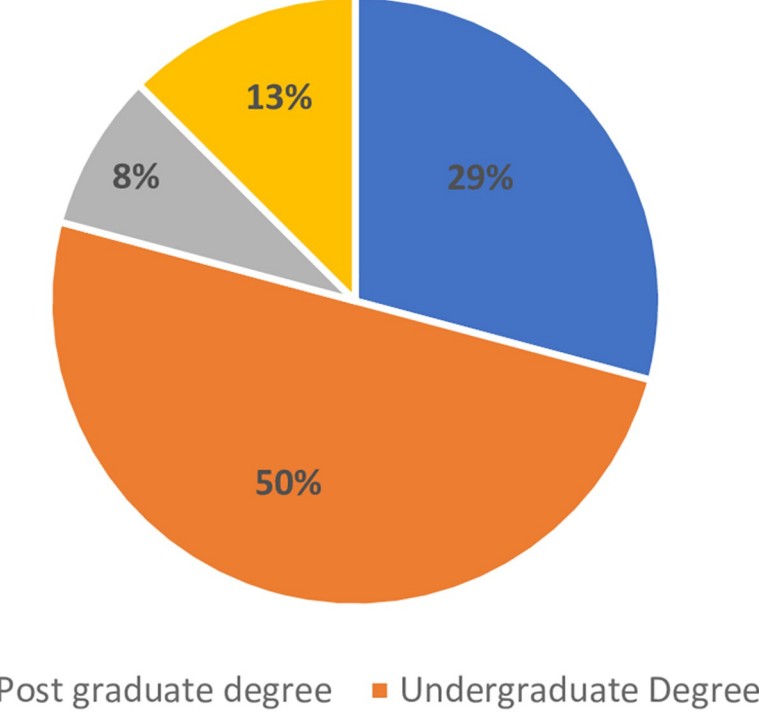

**Fig 1. Parental education level (n = 24).**

being pink was not *'remotely relevant'* to their infants or even to *'white people'*. Several HCPs before the interview did not consider how colour would be assessed in ethnic minority neonates. Assessing appearance in infants with darker skin pigmentation was difficult, requiring extra vigilance and additional training. Indeed, one mother believed her daughter's lower Apgar score was due to her darker skin tone, highlighting potential inaccuracies.

> *"She was fine, she came out and screamed and peed all over me and then started feeding within seconds. . . So she came out almost looking a little bit blue, but I think that's just because her skin tones a bit darker. So the pinkish hue was perhaps changed slightly."*
>
> (PA23- Black)

HCPs recognised the limitations of assessing neonatal colour, particularly in relying on *"pink all over. "*, some felt inaccurate for all babies, not just those from ethnic minorities. In response to their concerns about assessing skin colour, some HCPs assessed various body areas for colour, including the lips, mucous membranes, mouth, tongue, and eyelids. While some areas, such as mucous membranes, were considered quick reference points, the usefulness of peripheral areas, such as palms and soles, was acknowledged to be limited in the first few days after birth.

> *"We do say pink, but we just mean not blue."*
>
> (NNP04 –Mixed)

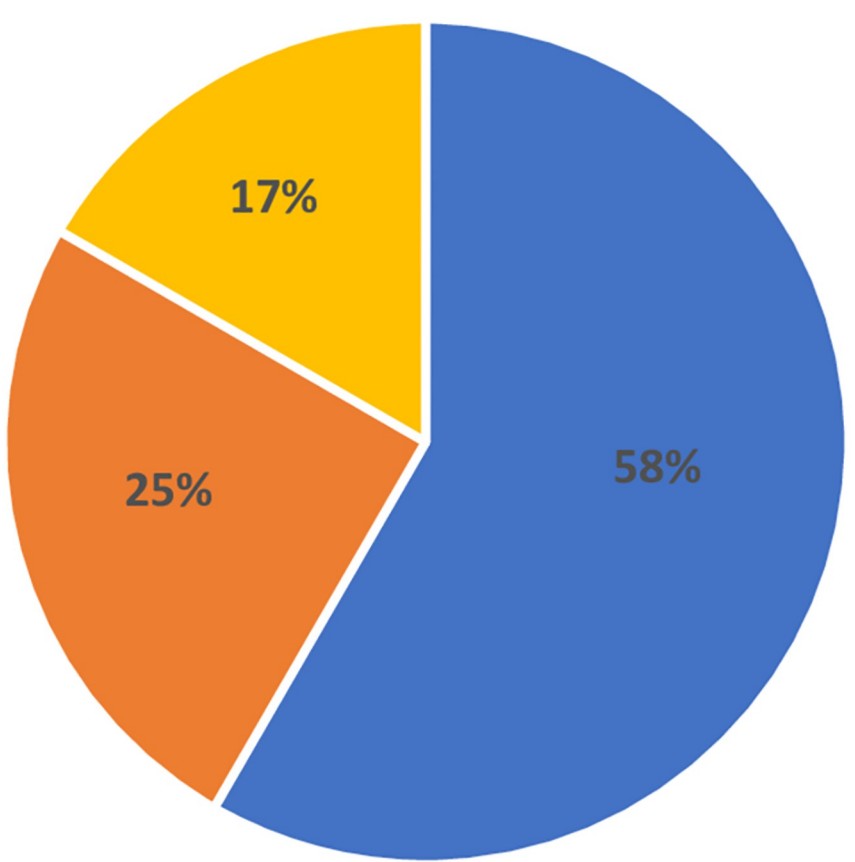

**Fig 2. Parental employment (n = 24).**

**Suggested changes to the Apgar score.** Only two HCPs believed that there was no need for alterations in the assessment of appearance within the Apgar score. However, one of these HCPs did not adhere to the textbook guidelines when they described assessing the appearance component of the Apgar score. The other professional did not feel sufficiently qualified to propose an alternative approach.

*"I think that one is looking at the colour of oxygenated blood. You aren't looking at skin colour. Yeah, well you are looking at skin colour and mucous membranes and fingernails and palms. But I don't think that the use of the word pink urm is discriminatory."*

(NNP07—White)

Several HCPs noted that changing the Apgar score could only be done if a purely objective and accurate assessment could be found. Other HCPs described several potential ideas, which are discussed below.

*Changes to the terminology of 'pink all over'.* Most HCPs expressed the need to change the use of the term "pink" skin, which was deemed racially insensitive and habitual. The multicultural nature of society was highlighted as a reason for the change, with suggestions for alternatives such as describing *"skin tone"* or using terms such as *"good"* or *"acceptable"*. Others suggested that "*pink*" could be used for mucous membranes rather than the skin or that the appearance component of the Apgar score could be completely scrapped. Other HCPs stressed the importance of assessing infant perfusion for well-being and suggested incorporating *"well perfused"* into the Apgar score. However, there is uncertainty among some HCPs about how to assess perfusion effectively and about its applicability to neonates with darker skin tones. While most HCPs advocated for reforming the Apgar score to reflect contemporary society, a few HCPs proposed abandoning the Apgar score altogether in favour of alternative assessment methods, questioning its relevance and accuracy. Alternative assessments suggested by HCPs included simpler assessments, such as ABC (airway, breathing, circulation), or monitoring of oxygen saturation through pulse oximetry.

*"I often rub out the pink and just put well perfused."*

(MW1- Mixed ethnicity)

*"Pink to me is a bit of an oxymoron, really. It's like, you don't get pink Black or Asian babies, do you?"*

(MW02 –Black)

Concerns were raised about resistance to change established practices among some colleagues, with examples of aggression or dismissiveness when questioning the use of *"pink"*.

*"I feel the Apgar score is kind of, . . . it's archaic now. . . the Apgar score was brought in for White European babies where you know, we're far from that now. We're a bit of a, you know, multicultural society now."*

(MW02—Black)

*Improved acknowledgement of and training in the assessment of ethnic minorities.* The current Apgar scoring system was inadequate for a multiethnic population. A separate scoring system was suggested for different ethnicities; however, the need for standardised Apgar scores worldwide has been recognised. Comprehensive training is essential for effectively assessing ethnic minority neonates, addressing the lack of confidence in this domain.

*"We have to change up the examination, so I think moving from this pathway that 'All neonates and all persons are examined in the same way'. I think we actually have to reassess that and say 'Oh maybe we should be doing things a bit differently for persons of darker skin colour'."*

(NNP09—Black)

## Cyanosis

**Detection of cyanosis.**   Most parents were uncertain about recognising signs of low oxygen in their children, with most never being concerned about their baby's oxygen levels. When concerns arose, they were from specific incidents such as oxygen use in the neonatal unit, multiple desaturations, or rapid breathing due to COVID-19. For example, the mother whose baby had been COVID-19-positive received prompt medical attention during which she was shown videos on recognising signs of respiratory distress. Another mother had received guidance on managing desaturations during feeding and received first-aid training before her child's discharge from the neonatal unit. One parent who had a family history of a baby dying was also particularly vigilant.

Parents who talked about recognising low oxygen levels in their baby suggested that they would observe their baby's skin colour, with some watching blue tones, especially on the face, or observing blue lips. However, one mother recounted her baby's face turning red while choking. If any blue appeared, the mothers stated that they would contact a doctor. Some parents also indicated that they would monitor the baby for breathing issues. One parent also mistakenly associated a rash with oxygen deficiency.

*"I assumed they would go slightly blue, but of course, that also might be what happens with White babies and I don't know what happens with mixed or Black babies."*

(PA22 –Mixed)

HCPs often focus on specific areas, such as the abdomen, extremities, or nailbeds, when checking for blue skin colour. However, they acknowledged that skin colour is not a reliable sole indicator of health, especially in unwell infants. HCPs were aware that factors such as ambient light could affect assessment accuracy. Additionally, detecting subtle changes in skin colour was particularly challenging for HCPs with colour blindness.

In addition to assessing the skin, HCPs also routinely assess the lips, mucous membranes, tongue colour, lower eyelid colour and gums, considering variations based on ethnicity. The capillaries of HCPs are frequently used to evaluate oxygenation and circulation. Both parents and HCPs considered neonatal responsiveness, tone, feeding adequacy, and contentment as indicators of well-being. Skin condition and hydration were also noted, with concerns that emollient use in Black neonates may mask a dull skin tone.

*"They go a little bit blue around the lips and but actually they go a little bit white the Black babies rather than blue that's it."*

(MW10 –White)

The HCPs assessed respiratory effort, including breathing rate, sternal recession, and grunting. They listened to their heart rate and checked for clubbing, with an emphasis on listening to parental concerns. The HCPs stressed the importance of pulse oximetry for detecting cyanosis, particularly in Black, Asian, and minority ethnic neonates. HCPs recommended routine use of pulse oximetry, but they acknowledged limitations and concerns about its accuracy in detecting darker skin tones.

**Challenges in identifying cyanosis in ethnic minority neonates.**   HCPs acknowledged challenges in detecting blue skin and lips in Black and minority ethnic neonates. The HCPs and parents held divergent views on the relevance of the term "*blue*" for nonwhite neonates. Some HCPs and parents acknowledged its importance; however, many expressed apprehensions about accurately detecting blue in Black, Asian, and minority ethnic neonates,

emphasising that a deeper blue hue may be seen. The difficulties in detecting cyanosis meant that several HCPs, particularly those from ethnic minority backgrounds, recounted instances where they identified cyanosis missed by their colleagues. This perceived disparity in recognition raised concerns about overlooked cases and potentially worse outcomes due to the lack of awareness about cyanosis detection in nonwhite neonates.

*"What happens is if people are not confident in detecting it, things you know you could get, you could have a really, really unwell baby that's been unwell for hours and nobody knows because 'ohh but the baby's colour looks fine'".*

(HV08 –Black)

**Training needs for detecting cyanosis.**    Only one midwife claimed adequate training in recognising cyanosis in an ethnic minority population. Many HCPs noted a lack of specific training in detecting cyanosis in Black, Asian, and minority ethnic neonates, despite receiving regular neonatal resuscitation updates. Hence, several HCPs felt underconfident in detecting cyanosis within this demographic, with one expressing uncertainty about what "*normality*" looks like. Perceived under confidence prompted one HCP to pursue personal learning. One HCP speculated that being Black herself might have provided her with a better understanding of infant assessment. HCPs desired training to determine the best way to assess cyanosis, and one parent urged improved training on recognising variations in skin tones.

*"I really don't think I've got a good, you know a good working knowledge of what I should be looking for in terms of skin colour."*

(HV03 –White)

Two HCPs questioned others about detecting cyanosis in Black, Asian and minority ethnic neonates. When one of the interviewees had undertaken training, she questioned an experienced trainer from the neonatal unit (NNU) who, in 10 years, had not considered how cyanosis detection may differ in those with darker skin. The other had been met with acknowledgement but then received no further training.

*""When I've been training with the neonatal teams and the fact is that they haven't even realized how limiting using colour as a sign of deterioration is, until we've had that conversation."*

*(MW04 –Asian)*

Some mentioned that recognising cyanosis in infants from ethnic minority backgrounds becomes easier with experience. One HCP noted that observing the colour change process in reverse as oxygen was administered to an ethnic minority infant aided her recognition.

## Jaundice

**Detection of jaundice.**    One-third of parents said they were not aware of signs of jaundice in their infant; however, six speculated they might notice it in their baby's skin. Two parents reported that they were unaware before guidance was given by HCPs in the hospital, and four were unaware until they had a child with jaundice. Two parents acknowledged awareness but provided no details, three gained awareness through a relative's experience of neonatal jaundice, and three stated awareness of skin colour before their child had jaundice. One mother

had read about jaundice, and another had awareness due to personal experience with a child with the condition.

> *"I didn't know that you had to look at the eyes and . . . the gums . . . the wee. I didn't know any of that until I'd actually looked it up myself. Yeah, nobody even in the hospital, no midwife, no support worker, no doctors, no nobody had pointed that out."*

(PA14—White)

HCPs and parents described that they would look at numerous signs to assess for jaundice. Parents commonly rely on observing their infant's skin colour to detect jaundice, with some noticing yellowing or darker-looking skin. The HCPs also assessed neonatal skin, with variations in recognising yellow hues, especially in Black and minority ethnic neonates. They looked for yellow in specific areas, such as the soles of the feet, palms, and nail beds, although they warranted caution, as visible jaundice may necessitate immediate blood tests. Comparing a baby's skin tone to that of its parents was suggested by a few HCPs, but some cautioned against assuming skin colour based on parental ethnicity. In addition to skin colour, HCPs stressed the importance of a comprehensive neonatal assessment for detecting jaundice, noting that it may not always be visible on the skin, especially in nonwhite neonates. They described checking the eyes and gums for yellowing while considering the infants' alertness, feeding pattern, and urine or stool colour. Misconceptions among parents and one HCP about jaundice symptoms existed; for instance, the baby would be vomiting or have a rash.

HCPs recommend conducting assessments in bright light and fully undressing the baby for a thorough evaluation, although challenges persist due to limited face-to-face appointments postpandemic.

> *"I'd probably look at his face in his eyes, his eyes, the whites of his eyes. But I guess then any of his skin I don't. But then I'm just, I'm guessing here I don't really know."*

(PA16—Asian)

**Challenges in detecting jaundice in neonates of black and minority ethnicities.** Several parents believed that identifying jaundice in their ethnic minority baby would not be more challenging. They mentioned focusing on other signs rather than relying solely on skin colour. These parents also felt that HCPs would not find jaundice detection harder in a black, Asian or minority ethnic baby because they perceived doctors to be experts. However, only one HCP felt that jaundice was easy to detect in all Black and minority ethnic neonates.

Other parents felt that their baby being from an ethnic minority would make parental jaundice identification more difficult, especially considering their lack of awareness regarding their child's "*normal*" skin colour at birth. As a result, one parent recounted struggling to answer a paediatrician's question about whether her child's skin colour looked "*normal*", as she did not yet know what "normal" would look like for her newborn. Many parents also felt that identifying jaundice may be more difficult for HCPs, especially if they are not accustomed to seeing jaundice in ethnic minority neonates. Within the parental interviews, three separate cases of jaundice were reported to have been missed at some point by at least one HCP.

> *"There was times when it was difficult to tell because of obviously the skin colour because ummm, yeah, because of the tone of my kids' skin tone, it was at times difficult to tell."*

(PA13—Asian)

Similarly, most HCPs felt that it was more difficult to recognise yellow hues in Black, Asian and minority ethnic neonates, particularly in children of dual heritage. Comparing a baby's skin tone to that of its parents was suggested by a few HCPs, but some cautioned against assuming a neonate's skin colour based on parental ethnicity. HCPs who felt jaundice were frequently missed in Black and minority ethnic neonates, with jaundice being at a higher level once identified; therefore, Black, Asian and minority ethnic neonates were overrepresented within readmissions to the NNU.

*"The most severe cases that are re-admitted and tend to be non-White. . . the ones that need to because they come to the unit and have multiple lights, tend not to be White."*

(NNP03—White)

Given that yellow skin colour is not always apparent with jaundice in Black, Asian, and minority ethnic neonates, HCPs emphasised the necessity for varied assessments and the importance of heightened vigilance. However, one parent felt that this elevated level of suspicion was absent, with too much emphasis placed on their baby's activity, meaning that other features of jaundice (yellow eyes, yellow skin and poor feeding) were ignored. This resulted in her child being over the exchange transfusion line by the time the jaundice was formally identified.

*"You can never be like 100% sure on OK, this baby looks yellow or it doesn't."*

*(MW10 –White)*

The identification of jaundice in Black and minority ethnic neonates was assisted when the HCPs themselves were from an ethnic minority background, where they identified jaundice that their colleagues from White backgrounds had missed. According to the parental interviews, one mother felt that their HCP had no problems identifying their child's jaundice, which may have been due to the HCP being from an ethnic minority background. However, in contrast, another parent reported that their child's jaundice was missed even by an ethnic minority HCP.

*"The midwife who came, also wasn't White, so maybe that's why she picked it up as well. I can't remember where she was from, but she had quite similar skin tone to me and maybe that's why she also noticed it."*

(PA22—Mixed)

The lack of timely identification was noted to increase the intensity of the treatment needed, such as needing exchange transfusion and potentially having a long-term impact on the infant's health. Failure to detect jaundice in the hospital setting was seen as demoralising for women, potentially leading to readmission and therefore potentially affecting bonding and breastfeeding. For one couple, the impact on breastfeeding and bonding was exemplified when a paediatrician came and took their child mid-feed to place him under phototherapy given the severity of their child's jaundice. The long-term impact of these delays in identification on mental health and trust in health services and HCPs was noted to be of concern. One father expressed distrust in health services due to his concerns that their care may have been inappropriate due to their ethnicity. Additionally, the couple whose child's severe jaundice was identified late was notified that the paediatrician had filed a complaint on their behalf. As a result of this complaint, the parents were informed that the hospital was going to put various measures

in place to prevent a similar incident from reoccurring, including buying bilirubinometers for the community and ensuring that the staff received annual training on jaundice. However, more than three years later, these measures were not implemented, resulting in a loss of trust in health services.

*"I was quite fearing, and I was trembling. Yeah, I was like did the doctor do something wrong or did the hospital do something wrong because of my ethnicity?"*

(PA11—Black)

**Addressing gaps in jaundice identification.**   HCPs cited a lack of adequate training in identifying jaundice among Black, Asian, and minority ethnic neonates, revealing a critical gap in awareness and proficiency. Limited formal instruction prompted reliance on informal learning from experienced colleagues, highlighting the need for structured training programs to address this deficit. Professionals recognised the necessity for enhanced training focusing on jaundice risk factors and bilirubin measurement techniques. Instances of missed cases underscore the importance of listening to parental concerns and integrating parental observations into clinical evaluations. HCPs often turn TCBs to screen for jaundice in darker-skinned neonates. While TCBs were seen as convenient and appreciated for being non-invasive, concerns were raised regarding their reliability and accuracy, particularly in ethnic minorities, with HCPs emphasising the need for standardised protocols and clearer guidelines. There are concerns that TCB inaccuracies lead to increased serum bilirubin test (SBR) results in ethnic minority neonates. By improving training and fostering a culture of vigilance, HCPs could enhance their ability to identify and manage jaundice effectively across all neonatal demographics, improving patient outcomes and reducing disparities in healthcare delivery.

*"Listen to the parents if the parents are worried, just do the bloods. Yeah, you know, if you've got a parent saying to you "I think my baby jaundice" I mean, they're probably right."*

(NNP03—White)

## Discussion

As part of a larger project, the aim of this study was to explore stakeholders' views of assessing the Apgar score, including the assessment of cyanosis, as well as the assessment of jaundice in ethnic minority neonates. Most parents felt that their infant's skin colour would not affect the assessments conducted on their infants, whereas HCPs felt that due to lack of training and exposure in practice skin colour assessments, neonates with darker skin tones would be affected.

One of the more significant findings to emerge from this study is that no HCP assessed the Apgar score entirely according to the textbook [1, 34].

Faults with an Apgar score have been discussed in numerous studies. O'Donnell (2006) [35] speculated that the Apgar score is measured retrospectively and has poor interrater reliability, which was confirmed in our interviews. Variations in Apgar scoring can also depend on location (community or hospital) [36, 39] and gestational age [37]. It has been suggested that the Apgar score should be determined uniformly and that the guidelines be followed strictly for it to be considered effective [38]. As acknowledged by the HCPs interviewed within

this study, further training is needed in the assessment of Apgar scores to ensure uniformity in scoring across all infants, particularly those from diverse races and ethnic backgrounds.

The Apgar score relies on the assumption that blue, pallor or pink will be displayed on the skin of all infants [9]. Recent studies have suggested that the use of the Apgar score introduces bias by lowering the score for black and brown-skinned infants [36, 39]. These lower Apgar scores may lead to additional interventions for neonates and possible admissions to the NICU [40]. HCPs rely upon the fact that if an infant is breathing, they should be the colour 'pink' [41]. However, other literature has questioned the relevance of the terminology 'pink all over' [11, 35], with this of further controversy in our interviews.

Yama & Marx (1991) [42] argued that the Apgar score should be calculated without the colour component. This opinion was reflected by some of the HCPs within the current study suggesting that the 'colour' component was the only part of the Apgar score that needed to be changed. Alternatively, some participants felt that the different parts of the Apgar score should be given different weights rather than each component remaining evenly weighted [43]. Other HCPs suggested a complete replacement for the Apgar score, but only if the assessment was objective and not susceptible to observer bias. Some Apgar score alternatives have been suggested within the literature to determine whether newborns need further care [44]. The neonatal resuscitation and adaptation score (NRAS) has been shown to correlate well with the Apgar score and may be more predictive of mortality [44]. The NRAS mutation eliminates the subjective elements of the Apgar score, which may disadvantage some infants [45, 46], particularly those with black or brown skin [47].

The difficulties in detecting cyanosis in neonates with darker skin tones, as highlighted in this study, emphasise the limitations of visual assessment. Guidelines and training resources by the Resuscitation Council (2011) [48] and Health Education England (2022) [49] indicated that colour alone is not a suitable method for evaluating oxygenation levels, which was echoed in our interview results.

HCPs suggested that assessment of the mouth and lips for cyanosis in all infants may be more effective. This finding is in line with the literature where tongue colour was shown to be a good indication of the need for supplemental oxygen at birth, regardless of ethnicity [12]. However, HCPs noted that the need for colour assessment has decreased with the widespread availability and use of pulse oximetry in clinical practice. Visual assessment of cyanosis should therefore be used only in the absence of pulse oximetry, particularly in any neonatal resuscitation scenario [48].

Nonetheless, the COVID-19 pandemic has highlighted disparities in the accuracy of pulse oximetry for black, Asian, or minority ethnic adults, indicating potential racial biases [17, 18, 50]. One recent study suggested that pulse oximetry may not be as accurate in preterm neonates of black and minority ethnicities with a greater incidence of occult hypoxemia [19]. Although pulse oximetry is more precise than visual inspection, the authors recommended avoiding saturations at the lower end of the normal range in preterm neonates of black ethnicity to minimise the risk of adverse outcomes in this population. Further investigations to explore the slight yet potentially significant differences in the functionality of pulse oximeters in neonates of various races and ethnicities are needed.

The difficulty of visually detecting jaundice in ethnic minority neonates reveals a critical gap in current screening practices. This study demonstrates that HCPs, while aware of these challenges, lack confidence in their assessments, however, the majority suggested that jaundice could be identified by assessing for yellow skin in other locations or by assessing the infant's general well-being. As a result, the adoption of more objective diagnostic tools was called for, such as TCBs. The widespread use of TCBs is suggested to reduce costs, nurse time and neonatal pain [51, 52]. However, HCPs identified cost concerns with extensive TCB use, particularly

in a community setting. However, it may be cost-effective compared to missed cases. In addition to human capital, compensatory harm caused by jaundice has cost the NHS approximately £150.5 million over the last decade [24, 53].

Nevertheless, the guidance proposed by the National Institute for Health and Care Excellence (NICE) in 2010 [54] recommends refraining from testing infants who lack visible jaundice. Consequently, there is a need for reconsideration of policy and further investigation to ensure a judicious approach toward jaundice assessment in neonates belonging to ethnic minorities. This is especially pertinent considering inaccuracies in diagnosis due to pigmentation [21, 22] and heightened susceptibility, findings substantiated by our interviews and supported by the literature [6].

Some parents reported changes in their infant's skin colour before HCPs, which has also been confirmed in the literature [29, 55–57]. However, gaps still exist in parental knowledge of jaundice globally [29, 58]. It would be pertinent to educate and equip parents about the detection of jaundice through other methods, such as looking at the sclera, gums, or eyes [50] which may significantly contribute to early intervention and improved infant health outcomes.

### Strengths and limitations

This is the first study to explore the experiences of a range of stakeholders in assessing or accessing care for jaundice, cyanosis and Apgar. By engaging with various individuals directly, this research provides a comprehensive understanding of the multifaceted dynamics at play. This inclusive approach enriches the depth of insights obtained and allows for a nuanced analysis of the factors influencing the phenomenon under examination. In addition to opening further investigation, this encourages ongoing dialogue and inquiry into critical issues within the field. To further the findings of this study, a larger-scale survey is recommended to gain a wider understanding of how the Apgar score is measured globally. It may also be important for an observational study to be conducted to assess how healthcare professionals assess infants in real time.

Study limitations need to be acknowledged. All participants who were interviewed spoke English fluently and did not require an interpreter; therefore, potentially interviews with high-risk groups may have been missed. However, the participation was open to all with steps being taken to reach out to the higher-risk groups, and the provision of an interpreter was proposed in the advert.

### Conclusion

In conclusion, this study revealed disparities in stakeholders' views on the Apgar score, emphasising the need for standardised training and policy adjustments for Black, Asian and other ethnic minority neonates. Challenges in cyanosis and jaundice detection emphasise the need for further investigation, particularly in enhancing the accuracy of existing tools for babies from various races and ethnic backgrounds. Listening to parents and ongoing education for healthcare professionals and parents, particularly in diverse populations, is of paramount importance to ensure that neonatal health disparities are reduced.

### Supporting information

**S1 File. Quotations.**
(DOCX)

## Acknowledgments

We would like to thank:

- All participants who kindly gave of their time to take part in the interviews and provide such valuable information.

- Yasmin Iqbal and Elham Kohta (Community engagement workers) and Josie Anderson (Policy, Research and Campaigns Manager, Bliss) for their support in participant recruitment.

- Zenab Barry (Director at National Maternity Voices, Strategic Adviser at NIHR Applied Research Collaboration (ARC) South London's Maternity and Perinatal Mental Health, Maternal Health Advocate) for her role in recruitment and debriefing participants.

## Author Contributions

**Conceptualization:** Frankie Fair, Gina Higginbottom, Sam Oddie, Hora Soltani.

**Data curation:** Frankie Fair, Amy Furness, Hora Soltani.

**Formal analysis:** Frankie Fair, Amy Furness, Hora Soltani.

**Funding acquisition:** Frankie Fair, Gina Higginbottom, Sam Oddie, Hora Soltani.

**Investigation:** Frankie Fair, Amy Furness, Hora Soltani.

**Methodology:** Frankie Fair, Amy Furness, Gina Higginbottom, Sam Oddie, Hora Soltani.

**Project administration:** Frankie Fair, Amy Furness.

**Supervision:** Hora Soltani.

**Validation:** Frankie Fair, Amy Furness.

**Visualization:** Frankie Fair, Amy Furness, Hora Soltani.

**Writing – original draft:** Frankie Fair, Amy Furness.

**Writing – review & editing:** Frankie Fair, Amy Furness, Gina Higginbottom, Sam Oddie, Hora Soltani.

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
