## [Decision Letter · Decision Letter 0]

18 Sep 2024

PONE-D-24-26223Understanding Stakeholder Perspectives on Apgar Score, Cyanosis and Identifying Jaundice in Ethnic Minority Neonates.PLOS ONE

Dear Dr. Soltani,

Thank you for submitting your manuscript to PLOS ONE. After careful consideration, we feel that it has merit but does not fully meet PLOS ONE’s publication criteria as it currently stands. Therefore, we invite you to submit a revised version of the manuscript that addresses the points raised during the review process.

**ACADEMIC EDITOR: **Please follow minor changes requested by the second reviewers,==============================

We look forward to receiving your revised manuscript.

Kind regards,

Stefan Grosek, Ph.D., M.D.,

Academic Editor

PLOS ONE

Journal Requirements:

“This research was commissioned by the NHS Race and Health Observatory.”

3. We noted in your submission details that a portion of your manuscript may have been presented or published elsewhere. [The interview results alone are published in the NHS Race & Health Observatory. Dual publication of interview results in a journal like PLOS ONE, even when the data has been published in a full report, is warranted for several compelling reasons. PLOS ONE's journal credibility ensures that the findings are subjected to rigorous peer review, adding an extra layer of scrutiny and validation that enhances the reliability and influence of the research. Furthermore, journal articles allow for targeted communication, enabling authors to tailor their findings to the specific interests and needs of PLOS ONE's readership. This targeted approach can highlight the most significant implications for practice, policy, or further research, making the results more relevant and actionable for the audience. Additionally, PLOS ONE's high visibility and impact due to its indexing in major databases and citation in subsequent research significantly increase the likelihood that the findings will be noticed, discussed, and built upon by other researchers. This enhanced visibility ensures that the research reaches a broader scientific community, contributing to the academic literature and supporting further advancements in the field.] Please clarify whether this [conference proceeding or publication] was peer-reviewed and formally published. If this work was previously peer-reviewed and published, in the cover letter please provide the reason that this work does not constitute dual publication and should be included in the current manuscript.

Additional Editor Comments:

Dear Authors

As was already said by bothe reviewers this article deals with a very interesting an important subject on understanding stakeholder perspevtives on Apgar score, cyansois and identifying jaundice in ethnic minority neonates.

After minor revision suggested by the second reviewer I believe this interesting article will be ready for acceptance.

Kind regards

Reviewers' comments:

Reviewer's Responses to Questions

**Comments to the Author**

1. Is the manuscript technically sound, and do the data support the conclusions?

Reviewer #1: Partly

Reviewer #2: Yes

2. Has the statistical analysis been performed appropriately and rigorously? 

Reviewer #1: N/A

Reviewer #2: N/A

3. Have the authors made all data underlying the findings in their manuscript fully available?

Reviewer #1: Yes

Reviewer #2: Yes

4. Is the manuscript presented in an intelligible fashion and written in standard English?

Reviewer #1: Yes

Reviewer #2: Yes

5. Review Comments to the Author

Reviewer #1: This paper raises some very important and socially relevant issues for today’s providers of neonatal healthcare. Although there is no real data presented, the authors have captured evidence of the inconsistencies in assigning Apgar scores and diagnosing jaundice for infants of color. This will hopefully serve as motivation for more systematic approaches and scientific solutions to these issues.

Reviewer #2: Dear authors,

you dealt with a very interesting an important subject and you seem have done a great analysis. I have some minor suggestions, though.

I do understand that the nature of the study is such that it needs to be described in great detail, I do think that maybe the manuscript is a bit too long and would benefit from being a bit more concise. Furthermore, there are several statements that appear both in the results as well as in the discussion section. I would suggest that discussion should be aimed more at explaining the reasoning behind certain opinions rather than repeating opinions themselves.

Finally, you mention that there are several limitations to the study and yet you name only one. I believe this should be addressed also.

Kind regards

6. PLOS authors have the option to publish the peer review history of their article (what does this mean?). If published, this will include your full peer review and any attached files.

Reviewer #1: No

Reviewer #2: **Yes: **Tina Perme, MD, PhD

---

## [Author Response · Author response to Decision Letter 0]

20 Sep 2024

Dear Members of the Editorial Board,

Thank you for providing feedback on our manuscript Understanding Stakeholder Perspectives on Apgar Score, Cyanosis and Identifying Jaundice in Ethnic Minority Neonates. We appreciate the reviewers' time and effort and have carefully considered each comment. Below, we provide a point-by-point response to address the suggestions and concerns raised.

Review 1: 

Response to reviewers:

Reviewer 1: 

We appreciate your thoughtful comments about our paper. 

Reviewer 2: 

Thank you for your helpful comments. We understand your concern about the length of the manuscript. However, given the complexity and depth of our study, we believe that the level of detail provided is essential to fully convey the results, and implications. We have, however, reviewed the manuscript, in particular the discussion to ensure that it remains as concise and focused as possible without compromising clarity. 

Sincerely,

Amy Furness

---

## [Editor Report · Decision Letter 1]

29 Sep 2024

Understanding Stakeholder Perspectives on Apgar Score, Cyanosis and Identifying Jaundice in Ethnic Minority Neonates.

PONE-D-24-26223R1

Dear Dr. Soltani,

We’re pleased to inform you that your manuscript has been judged scientifically suitable for publication and will be formally accepted for publication once it meets all outstanding technical requirements.

Kind regards,

Professor Stefan Grosek, Ph.D., M.D.,

Academic Editor

PLOS ONE

Additional Editor Comments (optional):

Dear Authors

Very nice and well written study on Understanding Stakeholder Perspectives on Apgar Score, Cyanosis and Identifying Jaundice in Ethnic Minority Neonates.

I propose to accept the article for publication

Kind regards
---

## [Editor Report · Acceptance letter]

4 Oct 2024

PONE-D-24-26223R1 

PLOS ONE

Dear Dr. Soltani, 

I'm pleased to inform you that your manuscript has been deemed suitable for publication in PLOS ONE. Congratulations! Your manuscript is now being handed over to our production team.

Kind regards, 

on behalf of

Professor Stefan Grosek 

Academic Editor

PLOS ONE